# Safety and Immunogenicity of Inactivated *Bacillus subtilis* Spores as a Heterologous Antibody Booster for COVID-19 Vaccines

**DOI:** 10.3390/vaccines10071014

**Published:** 2022-06-24

**Authors:** Johnny Chun-Chau Sung, Nelson Cheuk-Yin Lai, Kam-Chau Wu, Man-Chung Choi, Chloe Ho-Yi Ma, Jayman Lin, Cheong-Nang Kuok, Wai-Leng Leong, Weng-Kei Lam, Yusuf Khwaja Hamied, Dominic Man-Kit Lam, Eric Tung-Po Sze, Keith Wai-Yeung Kwong

**Affiliations:** 1Research Department, DreamTec Cytokines Limited, Hong Kong, China; johnnysung@dreamtec.hk (J.C.-C.S.); nelsonlai@dreamtec.hk (N.C.-Y.L.); kcwu@dreamtec.hk (K.-C.W.); jerrychoi@dreamtec.hk (M.-C.C.); chloema@dreamtec.hk (C.H.-Y.M.); info@dreamtec.hk (J.L.); 2Oristry BioTech (HK) Limited, Hong Kong, China; 3Meserna Therapeutic (HK) Limited, Hong Kong, China; 4L&L Immunotherapy Company Limited, Hong Kong, China; dlam@worldeye.org; 5Macao Greater Bay Area Association of Healthcare Providers, Macau 999078, China; tomas.kuok@gbahealth.org (C.-N.K.); vian.leong@gbahealth.org (W.-L.L.); vick.lam@gbahealth.org (W.-K.L.); 6Cipla Limited, Mumbai 400013, India; ykh@cipla.com; 7Torsten Wiesel International Research Institute, Sichuan University, Chengdu 610064, China; 8School of Science and Technology, Hong Kong Metropolitan University, Hong Kong, China; esze@hkmu.edu.hk

**Keywords:** SARS-CoV-2, COVID-19, oral vaccine, *Bacillus subtilis*, spike protein, sporulation

## Abstract

The coronavirus diseases 2019 (COVID-19) pandemic caused by severe acute respiratory syndrome coronavirus 2 (SARS-CoV-2) infections have threatened the world for more than 2 years. Multiple vaccine candidates have been developed and approved for emergency use by specific markets, but multiple doses are required to maintain the antibody level. Preliminary safety and immunogenicity data about an oral dose vaccine candidate using recombinant *Bacillus subtilis* in healthy adults were reported previously from an investigator-initiated trial in Hong Kong. Additional data are required in order to demonstrate the safety and efficacy of the candidate as a heterologous booster in vaccinated recipients. In an ongoing, placebo-controlled, observer-blinded, fixed dose, investigator-initiated trial conducted in the Macau, we randomly assigned healthy adults, 21 to 62 years of age to receive either placebo or a *Bacillus subtilis* oral dose vaccine candidate, which expressed the spike protein receptor binding domain of SARS-CoV-2 on the spore surface. The primary outcome was safety (e.g., local and systemic reactions and adverse events); immunogenicity was a secondary outcome. For both the active vaccine and placebo, participants received three courses in three consecutive days. A total of 16 participants underwent randomization: 9 participants received vaccine and 7 received placebo. No observable local or systemic side-effect was reported. In both younger and older adults receiving placebo, the neutralizing antibody levels were gradually declining, whereas the participants receiving the antibody booster showed an increase in neutralizing antibody level.

## 1. Introduction

Since the first cases of coronavirus disease 2019 (COVID-19) were revealed in December 2019, there have been hundreds of millions of infections worldwide [1]. Vaccination is the most effective means of controlling the spread of the disease and death. Since 2020, there were pressures to develop safe and effective vaccines to fight the virus, and at least 141 vaccine candidates were developed and tested to some degree [2]. Although more than 65.1% of the world’s population received at least one dose of a vaccine, variants such as B.1.617.2 (delta) and B.1.1.529 (omicron) penetrated in many countries and infected even people that were fully vaccinated [3]. The transmission of these variants increased the risk to the elderly and persons with pre-existing medical conditions and cause severe disease, hospitalization and death [4]. Use of booster vaccines is the most effective way to alleviate waning immunity against severe acute respiratory syndrome coronavirus 2 (SAR-CoV-2) and its variants. For example, a third dose of 30 μg homologous mRNA vaccine BNT162b2 (Pfizer–BioNTech) 7.9 to 8.8 months after the primary two-dose series was found to increase neutralizing antibody titer against the delta variant by a factor of 5 or 2.1, respectively [5].

Although COVID-19 vaccines can effectively provide protection against COVID-19, there has been a worldwide decline in acceptance towards COVID-19 vaccines and booster doses [6]. Serious side-effects, safety concerns and the fear of needles are some of the major reasons behind the hesitancy toward COVID-19 vaccines [7,8]. While the safety concerns can be alleviated through vigorous testing, a needle-free vaccine booster against COVID-19 provides an irreplaceable advantage over injected vaccines. Despite challenges faced during the development of a novel vaccine other than through injection route, oral, nasal or dermal patches delivery of vaccines have been developed against various diseases [9]. The non-pathogenic status of *B. subtilis*, with a safe record of human and animal use as both a probiotic and a food additive [10], has made *B. subtilis* an exceptional candidate for platforming novel oral vaccines. The simplicity of the construction of recombinant spores presenting heterologous protein, combined with the easiness of spores’ production and administration, make them especially interesting carriers of antigens in mucosal vaccines [11]. The *B. subtilis* spore-based vaccines have also been shown to stimulate both systemic and localized immune responses with balanced Th1/Th2 polarization for optimal antibody production [12]. Immunogenic adjuvants are compounds that enhance the potency, quality or longevity of specific immune responses. Interestingly, *B. subtilis* has not only shown itself to be an immunogenic adjuvant in various vaccines against different bacterial and viral pathogens, but also can express genetically engineered antigens to be on the outer surfaces of its spores [11,13,14,15]. *B. subtilis* endospores can therefore serve as a platform for presentation of antigens and as an immunogenic adjuvant.

Another obstacle of vaccines is distribution. As of May 2022, there are still many countries with under 10% vaccination coverage according to WHO’s statistics [16]. Although the supply of vaccines being unequal is one of the reasons for the low vaccination rate, the major reason is the transportation and storage of vaccine. For instance, the Pfizer–BioNTech vaccine must be stored at −80 °C, and the Moderna vaccine has to be stored at −20 °C. A rigid requirement for cold-chain transport greatly increases the difficulties for global distribution [17]. Other than vaccine storage and transportation, the efficacies are another focus. Many researchers suggested vaccine efficacy declined 6 months after the second dose of injection, especially towards new variants, indicating a booster dose may be required every year [18,19]. This does not only cause a tremendous economic disadvantage but also adds to the workload of medical professionals. On the other hand, a *Bacillus* oral vaccine would be a solution to the instability and also ease the economic difficulties of current vaccines. Mucosal immunization using engineered microorganisms with antigens expressed on their outer surfaces by oral intake is a convenient way of providing booster vaccines.

In response to the need for vaccine boosters, a coordinated program for an oral dose, recombinant *B. subtilis*-spore-based COVID-19 vaccine candidate was conducted in Hong Kong [20]. The program was designed to test the safety and immunogenicity of the vaccine candidate, where three courses of 5 × 10^7^ spores/kg in healthy adults at 48 to 72 years of age within 28 days elicited an acceptable level of reactogenicity. In addition, no adverse event nor side-effects were observed. The vaccine candidate demonstrated both safety and efficacy as a first dose vaccine, but additional data are required in order to extend its usage as a heterologous (different from the primary vaccine) booster candidate.

In this study, the safety and immunogenicity data from an investigator-initiated trial of an ongoing randomized, placebo-controlled, observer-blinded, fixed dose in Macau are reported to evaluate the vaccine candidate as a third dose booster. These data include evaluation of dose level at in adults 21 to 62 years of age.

## 2. Materials and Methods

### 2.1. Trial Objectives, Participants and Oversight

Healthy adults 18 to 65 years of age who were previously administrated with 2 doses of vaccine of any brand (either BNT162b2 [21] or BBIBP [22] COVID-19 vaccine was available in Macau) at least 6 months ago were eligible for inclusion. Key exclusion criteria were known infection with SARS-CoV-2 virus, human immunodeficiency virus, hepatitis C virus or hepatitis B virus; an immunocompromising condition; a history of autoimmune disease; a previous clinical diagnosis of COVID-19; the receipt of medications intended to prevent COVID-19; and a positive nasal-swab result on a SARS-CoV-2 nucleic acid amplification test within 24 h before the receipt of trial booster or placebo.

### 2.2. Trial Procedures

The trial was conducted according to relevant national and international guidelines. Written consent was obtained from all participants. The protocols used were reviewed and approved by Doctors Think Tank Academy Ethic Committees. Sixteen (16) trial participants from 21–62 years old were randomly assigned into the active booster or placebo groups. The booster was prepared as previously described [20]. Briefly, *B. subtilis* expressing the spike protein receptor binding domain from SARS-CoV2 (Wuhan-Hu-1, Clade 19, a.a. 319–541, UniProt ID: P0DTC2 [23]), linked with the spore coat protein CotC, were cultured and induced into spores. The spores were then killed in an oven at 120 °C for 240 min to achieve overkill before subjected to 5 freeze drying cycles [24,25]. The freeze-dried antibody boosters were then counted with a hemocytometer and encapsulated with sodium alginate in enteric coated capsules.

Participants in active booster group were orally administrated with an enteric coated capsule containing 5 × 10^10^ inactivated *B. subtilis* spores encapsulated in sodium alginate microspheres on days 1, 2 and 3 (3 courses). Participants in the placebo group were assigned to receive a placebo capsule containing wild-type *B. subtilis* spores with sodium alginate microspheres, administered in the same schedule as the vaccine group. All participants were observed for 1 h after the administration of booster or placebo to identify any immediate adverse events. Furthermore, 5 mL blood samples were drawn on days 0, 14, 35 and 49 for safety and immunogenicity assessments; serum of each blood sample was collected and stored at −80 °C until assay. The titer of neutralizing antibodies was determined with a CLIA-based assay MAGLUMI SARS-CoV-2 neutralizing antibody detection kit (Snibe Diagnostic, Shenzhen, China).

### 2.3. Safety

During the blood sampling period, primary safety end points were assessed, including not only local reactions and systemic events, but also unsolicited adverse events and serious adverse events. Participants could apply the safety stopping rules anytime during the administration of vaccine or placebo.

### 2.4. Statistical Analysis

Data are expressed as the mean ± standard error of the mean (SEM), and statistical significance was determined by student’s *t*-test, and one-way or two-way ANOVA with Tukey’s *post hoc* test in GraphPad Prism 7.1 (San Diego, CA, USA). Shapiro–Wilk tests were performed to ensure normality of sample distribution. Data were considered significantly different when the *p*-value was less than 0.05.

## 3. Results

### 3.1. Characteristics of the Participants

On 12 February 2022 (day 1), a screening was conducted on a total of 18 healthy adults (men and nonpregnant women) at the site in Macau. Then, 16 participants were randomly assigned to the booster or placebo groups: 9 participants received the booster and 7 received the placebo. All participants had received two doses of vaccine (five received BNT162b2 and 11 received BBIBP) at least 6 months before this booster study (average, 8.31 ± 1.92 months). The ratios of vaccination brands (BNT162b2:BBIBP) in the booster and placebo groups were 3:6 and 2:5 respectively. The average time durations between the second dose and this booster study in the booster and placebo groups were 8.67 ± 2.00 and 7.86 ± 1.86 months, respectively, leaving no significant difference between the two groups (two-sample *t*-test, *p* > 0.05). Several studies reported that the neutralizing antibody level declined over time and reached a stabilization phase at low levels (~60% for BNT162b2 and 15% for BBIBP), from 6 months after the second dose of the primary vaccination cycle [26]. The starting date of the booster study was thus chosen using at least 6 months as the criterion. The average age of all participants was 39.8 ± 11.8 years old. The average ages for the booster and placebo were 39.0 ± 12.5 and 38.7 ± 12.4 years old respectively. Again, there was no significant difference between the two groups (two-sample *t*-test, *p* > 0.05).

### 3.2. Safety

Participants in all ages who received 5 × 10^10^ inactivated *B. subtilis* spores did not report any local reaction, not even any mild abdomen pain, muscle pain, joint pain, headache or diarrhea, within 7 days after finishing all three courses. Since the vaccination was administrated through the oral route, there was no injection site, and thus no common effects such as injection site pain, redness or swelling, as for the other injection-type vaccinations.

No fever nor chills were reported from participants of any age who received 5 × 10^10^ inactivated *B. subtilis* spores, which are common side-effects for current COVID-19 vaccines. No severe systemic event was reported in the booster group or placebo group. No serious adverse event was reported, and no participant met the criteria of stopping rules to cease the administration of the vaccine candidate.

### 3.3. Immunoreactivity

Before administration of the booster or placebo, blood samples were collected on days 1 and 15 to measure the neutralizing antibody level against SARS-CoV-2. Table 1 summarizes the level of neutralizing antibody (μg/mL) of each individual for both booster and placebo treatment groups. The level neutralizing antibody on day 1 varied tremendously, from 0.036 to 1.030 μg/mL. In general, neutralizing antibody levels from participants who previously received BNT162b2 were found to be higher than those who had taken BBIBP.

For ease of analysis of data with large discrepancies, the levels of neutralizing antibody on days 15, 36 and 50 are expressed by normalization with the levels on day 1 (by assuming those to be 1). Table 2 illustrates the normalized level of neutralizing antibody for both groups over time. The neutralizing antibody level was normalized by expressing it as a relative level to day 1 for each individual. Table 3 shows the average levels of normalized neutralizing antibody for the booster and placebo treatment groups on days 15, 36 and 50. The data were treated separately according to the vaccinations taken (i.e., BNT162b2 and BBIBP) previously by each participant and in a combined way (i.e., overall).

Some studies revealed the levels of neutralizing antibodies against SARS-CoV-2 gradually declined after vaccination with BNT162b2 or BBIBP [18,27,28]. This decline in neutralizing antibody was also observed on day 15 in most of the participants (except Booster #3) when compared with the levels in day 1. As shown in Table 3, the average level of normalized neutralizing antibody in the booster (BNT162b2) group was found to be significant different from that of the booster (BBIBP) group on day 15. Otherwise, there was no significant difference among other data (one-way ANOVA, *p* > 0.05) on the same day. The significant difference may have been caused by the small sample size, since no significant difference was found when considering all participants that were previously treated with BNT162b2 (0.935 ± 0.081) or BBIBP (0.732 ± 0.109) on day 15. To minimize the bias due to insufficient sample size, both booster and placebo groups were considered by combining all data within the same treatment (i.e., overall). By following one phase exponential decay model, the estimated half-lives (*t*1/2) of the neutralizing antibody of the vaccine (overall) and placebo (overall) groups were found to be 36.9 and 51.4 days, respectively.

After administration of the booster and placebo on days 15–17, blood samples were collected on days 36 and 50 to measure the neutralizing antibody levels (Table 1). As shown in Table 3, there was no significant difference for data within the same treatment group on days 36 and 50 (one-way ANOVA, *p* > 0.05). Therefore, booster (overall) and placebo (overall) were used in the assessment, and a plot showing the average levels of neutralizing antibody for the two treatments over time was created (Figure 1). After boosting, an upward trend for the normalized neutralizing antibody levels in the booster groups were observed on days 36 and 50 (Figure 1). The placebo patients who did not receive a booster continued to show a decline. By analysis of the data by 2-way ANOVA with Tukey’s post hoc test (Table 4), we observed significant differences for the factors “time,” “time x treatment” and “subject” (*p* < 0.05); there was no significant difference for the variation caused by “treatment” only. (*p* > 0.05). On day 50 (35 days after boosting), the normalized neutralizing antibody of the booster group was increased to an average level not significantly different to that of day 1 (one-sample *t*-test, *p* > 0.05).

## 4. Discussion

Although heterologous vaccination delivery strategies have been demonstrated successfully in many different infectious diseases [20], before the COVID-19 pandemic, they were usually inactivated virus vaccines and recombinant protein subunits only. This is the first safety and immunoreactivity report of using an oral dose of *B. subtilis* bacterial spores as a heterologous COVID-19 vaccine booster candidate for use in healthy adults who have previously received two doses of either mRNA or inactivated virus vaccines. The oral dose vaccine booster candidate in this study is made by a recombinant technique, allowing expression of the spike protein receptor binding domain of SARS-CoV-2 on the surfaces of *B. subtilis*, a non-pathogenic bacterium [29]. The oral booster vaccine candidate was found to be immunogenic in all participants of the vaccine group, regardless of which primary vaccine regimen they had received, and the results were in general in agreement with previous results of the same oral vaccine candidate after the first dose [20]. Apart from enhancing the neutralizing antibody titer in the host, previous study of the vaccine candidate by incubation of *B. subtilis* spores with immune cells detected upregulation of in vitro cytokine levels [20]: significant increases in IgM, IgG and IgA antibodies against the spike protein receptor binding domain were detected in the mouse serum and intestinal tissue.

Results from the study support not only the use of inactivated *B. subtilis* spores to boost the neutralization antibody levels of subjects, but also that the effect lasts over a few weeks after administration of the booster candidate. The direct effects of the inactivated oral vaccine booster candidate are expected to activate immune responses at the level of the gut mucosa solely by the surface of the spore, followed by triggering systemic effects [30]. Similar to this study, Maeda et al. [31] successfully increased the production of interferons by the use of heat-killed *Lactobacillus plantarum* L-137 in rodent studies to protect against influenza virus infection.

Although a study reported that systemic side-effects for a heterologous vaccination strategy using viral vectored and mRNA type vaccines as boosters were more common than by homologous vaccination [15], surprisingly, there was no local reaction nor systemic event observed by the use the inactivated bacterial spores as a heterologous vaccine booster in this study. As an oral dose will not introduce any wound during vaccination, some commonly found local side-effects of injection-type vaccinations, such as tenderness and pain around the injection site, will not exist. It can be a booster option to those patients who had adverse reactions to previous injections. In addition, inactivated bacterial spores will not induce any infectious diseases themselves, as they could not replicate nor secrete toxins, which is suitable for patients with severely weakened immune systems [16]. The intestinal flora forms a tight community in the intestine and are essential in the formation of gut-associated lymphoid tissues (GALT), which are responsible for the mucosal immune response. A superantigen found on *B. subtilis* spores but not on vegetative cells is required for the formation of GALT [32,33]. This stimulation of GALT formation is likely beneficial to the host, providing important intestinal functions and diversifying the antibody repertoire [34,35]. While if the *B. subtilis* spores booster can stimulate GALT formation and bring beneficial effects in the current study, we observed no adverse effect in the participants administrated with the killed spores. *B. subtilis* has long been used as a probiotic bacterium to modulate intestinal flora [36]. Recent reports have also shown that *B. subtilis* can also enhance the intestinal barrier and has great potential for developing antiviral agents [37,38]. Nevertheless, COVID-19 is infamous for inducing a cytokine storm in severe cases, which releases large amounts of pro-inflammatory cytokines, and this is likely to be brought on by the superantigen property of SARS-CoV-2 antigens [39]. Considering there is no discomfort nor other adverse effects experienced by the participants, the chance of inducing a cytokine storm with the reported vaccine booster is minimal.

The good stability of inactivated *B. subtilis* spores at room temperature has an advantage for the distribution of vaccine [17]. The platform using *B. subtilis* spores displaying antigen on their surfaces is suitable for allocation to areas with insufficient refrigeration facilities. Inactivated *B. subtilis* spores in a capsule form can support a much longer shelf life under ambient conditions than injection-type vaccines, and even longer than that of freeze-dried live bacterial spores. One of the major drawbacks of oral dose vaccines is the high doses required [16]. An optimal dose for the *B. subtilis* spores providing adequate protection against COVID-19 is yet to be examined. Due to several constrains, this study only checked on the participants for two months. The duration of the antibody-boosting effect is therefore yet to be discovered. Aside from the serum neutralizing antibody level, other assays, such as the SARS-CoV-2 pseudovirus neutralization assay, could also be performed in further studies investigating the boosting effect of the reported booster. Future work on inactivated, oral dose, recombinant *B. subtilis* bacterial spores should include a larger scale clinical study to examine dose level and efficacy, and any other biological activities the inactivated spores trigger in the host.

## 5. Patents

Patents resulting from the work reported in this manuscript were filed: Chinese (patent number: 202111143384.9), Hong Kong (patent number: 32021042343.2) and PCT.

## Figures and Tables

**Figure 1 vaccines-10-01014-f001:**
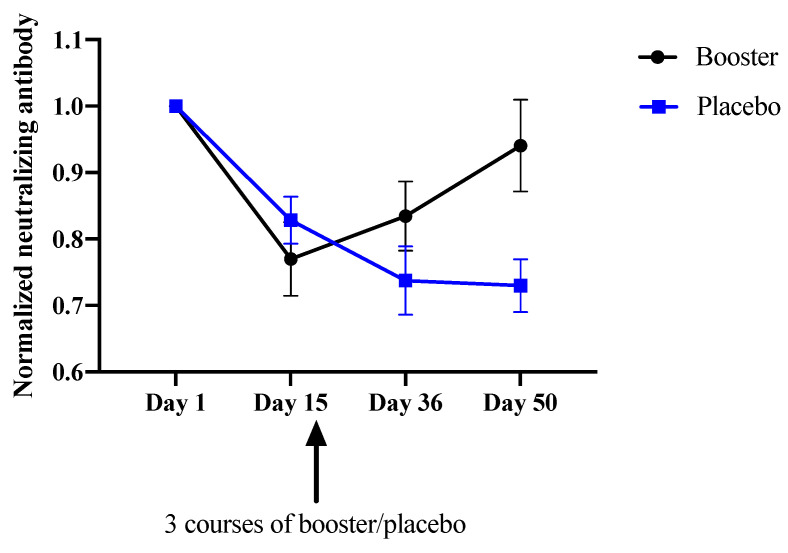
Normalized levels of the neutralizing antibody in both booster and placebo groups over time.

**Table 1 vaccines-10-01014-t001:** Levels of neutralizing antibody in all participants over time.

Treatment	Previous Vaccination Received	Level of Neutralizing Antibody (μg/mL)			
		Day 1	Day 15	Day 36	Day 50
Booster #1	BNT162b2	1.030	0.907	0.900	0.929
Booster #2	BNT162b2	0.731	0.682	0.687	0.678
Booster #3	BNT162b2	0.537	0.576	0.601	0.566
Booster #4	BBIBP	0.119	0.082	0.099	0.115
Booster #5	BBIBP	0.045	0.030	0.030	0.034
Booster #6	BBIBP	0.077	0.047	0.062	0.063
Booster #7	BBIBP	0.047	0.030	0.030	0.066
Booster #8	BBIBP	0.071	0.043	0.049	0.049
Booster #9	BBIBP	0.036	0.030	0.034	0.034
Placebo #1	BNT162b2	0.825	0.719	0.698	0.608
Placebo #2	BNT162b2	0.542	0.498	0.459	0.461
Placebo #3	BBIBP	0.284	0.233	0.248	0.246
Placebo #4	BBIBP	0.040	0.034	0.032	0.030
Placebo #5	BBIBP	0.103	0.080	0.055	0.067
Placebo #6	BBIBP	0.492	0.448	0.320	0.334
Placebo #7	BBIBP	0.108	0.070	0.066	0.062

**Table 2 vaccines-10-01014-t002:** Normalized levels of neutralizing antibody for all participants over time.

Treatment	Previous Vaccination Received	Day 15	Day 36	Day 50
Booster #1	BNT162b2	0.881	0.874	0.902
Booster #2	BNT162b2	0.933	0.940	0.927
Booster #3	BNT162b2	1.073	1.119	1.054
Booster #4	BBIBP	0.689	0.832	0.966
Booster #5	BBIBP	0.667	0.667	0.756
Booster #6	BBIBP	0.610	0.805	0.818
Booster #7	BBIBP	0.638	0.638	1.404
Booster #8	BBIBP	0.606	0.690	0.690
Booster #9	BBIBP	0.833	0.944	0.944
Placebo #1	BNT162b2	0.872	0.846	0.737
Placebo #2	BNT162b2	0.919	0.847	0.851
Placebo #3	BBIBP	0.820	0.873	0.866
Placebo #4	BBIBP	0.850	0.800	0.750
Placebo #5	BBIBP	0.777	0.534	0.650
Placebo #6	BBIBP	0.911	0.650	0.679
Placebo #7	BBIBP	0.648	0.611	0.574

**Table 3 vaccines-10-01014-t003:** Average of normalized levels of neutralizing antibody for each group over time (normalized by assuming the level of neutralizing antibody level on day 1 as 1).

	Previous Vaccination Received	Day 15	Day 36	Day 50
Booster Group	BNT162b2	0.962 ± 0.099 *	0.978 ± 0.127	0.961 ± 0.081
	BBIBP	0.674 ± 0.084 *	0.801 ± 0.098	0.930 ± 0.256
	Overall	0.770 ± 0.166	0.834 ± 0.156	0.940 ± 0.207
Placebo Group	BNT162b2	0.895 ± 0.033	0.846 ± 0.001	0.794 ± 0.080
	BBIBP	0.801 ± 0.098	0.694 ± 0.139	0.704 ± 0.110
	Overall	0.828 ± 0.094	0.737 ± 0.136	0.730 ± 0.105

* Significant difference between booster (BNT162b2) and booster (BBIBP) groups (two-sample *t*-test, *p* < 0.05).

**Table 4 vaccines-10-01014-t004:** Two-way ANOVA with Tukey’s post hoc test results for the data of participants’ normalized neutralization antibody levels at different times (before and after treatment).

Source of Variation	% of Total Variation	*p* Value	Significance
Time × Treatment	9.807	0.0108	*
Time	27.68	0.0001	***
Treatment	3.673	0.1861	not significant
Subject	26.60	0.0127	*

* Significant for *p* < 0.05; *** significant for *p* < 0.001.

## Data Availability

The data presented in this study are available on request from the corresponding author. The data are not publicly available due to privacy issue.

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
