# Peer review of "Safety and Immunogenicity of Inactivated Bacillus subtilis Spores as a Heterologous Antibody Booster for COVID-19 Vaccines"

_vaccines, 2022, doi:10.3390/vaccines10071014_

Round 1

Reviewer 1 Report

The study is valuable in considering the use of bacillus spores as a potential vaccine carrier. 

However there are perhaps some areas that can be improved. 

While the participant pool is small, appreciate the difficulty, however there are significant details lacking

The expression of the spike on the Bacillus, is it the whole spike? Which spike? Delta? Omicron? No info in M&M

Intro can cover info on the use of Bacillus in vacines 

Evidence of spike expression on spore? Why isnt the control group immunized with Bacillus without spike? 

Day 7 blood not collected? Why not? 

How were the spore counts performed? M&M details required.

How were the antibody titers normalized? To pre vaccination?

The above points could be clarified and addressed before a proper evaluation. 

Reviewer 2 Report

I read with great interest the manuscript. I find the concept and study design interesting. The manuscript is written well and easy to follow. However some major issues need to be addressed before the publication.

The trial procedure under Materials and Methods is not clear. There is no information in this part that B. subtilis was coated with SARS-CoV-2 spike proteins. This data is explained under discussion.

As B. subtilis can function as superantigen and indirectly activate B lymphocytes has been already primed with previously administered SARS-Cov-2 vaccines in tested individuals, authors must comment on this issues and explain/address this issue in the manuscript.

Author Response

We have updated the Materials and Methods section as well as discussed in the discussion part. Please see the revised and updated version of manuscript.

Reviewer 3 Report

Review of Sung, et al.

Sung, et al.  report a small study of 17 humans where 9 received a vaccine for SARS-CoV-2 and 7 received a placebo.  The vaccine is a B. subtilis oral vaccine strain expressing the SARS-CoV-2 spike protein. They examined the safety of this spore vaccine candidate.  The inactivated spore vaccine doesn’t require refrigeration and is oral.   In their previous study, they tested freeze dried live recombinant B. subtilis spores.

MAJOR

Page 2:  “japed” is not appropriate and needs to be removed

Author Response

The text has been revised in the revised manuscript.

Reviewer 4 Report

Article describing the immune response and secondary effects to a Bacillus subtilis spore oral vaccine expressing the spike 28 protein receptor-binding domain of SARS-CoV-2 on the spore surface. The article is well written in a comprehensible way.

 The authors have already described the activity of this vaccine in an article in the same Vaccines Journal (Sung, J.C.C.; et al. Expression of SARS-CoV-2 Spike Protein Receptor Binding Domain on Recombinant B. subtilis on Spore Surface: A Potential COVID-19 Oral Vaccine Candidate. Vaccines 2022, 10, 2. https:// doi.org/10.3390/vaccines10010002).

My main concern has to do with the way of measuring the antibody (nAb) levels produced by the booster heterologous vaccination. nAb data were expressed by normalization with Day 1 (line 144). This is a strange way of giving the results of immunoreactivity. Measurements should be an objective measure, not a relative one.

For instance, were nAb units the same for both vaccines (BNT and BBIBP) on day 1? Were the nAB units the same for both groups vaccine/placebo on day 1? In their previous work, the authors expressed Ab levels as AU (CLIA assay) and with nAb measured in a pseudovirus neutralizing assay.

On day 50 …. How can be a booster effect be considered if there were less antibodies than before the booster dose? (line 170).

In other works, the effect of booster vaccination has been measured by quantitative antibody titling and by neutralizing assays. In these works, antibody levels after vaccination were always higher than at the moment of vaccination, showing a clear booster effect, not as in this work.

In summary, the authors proved that people of the vaccinated group produced more nAb after aproximately 35 days post booster doses than the placebo group, but they didn’t show an increase in the nAb titers after that time in the vaccinated group.

Atmar RL, et al. Homologous and Heterologous Covid-19 Booster Vaccinations. N Engl J Med. 2022 Mar 17;386(11):1046-1057.

Alidjinou EK, et al. Immunogenicity of BNT162b2 vaccine booster against SARS-CoV-2 Delta and Omicron variants in nursing home residents: A prospective observational study in older adults aged from 68 to 98 years. Lancet Reg Health Eur. 2022 Jun;17:100385. doi: 10.1016/j.lanepe.2022.100385. 

Romero-Ibarguengoitia ME, et al. Effect of the third dose of BNT162b2 vaccine on quantitative SARS-CoV-2 spike 1-2 IgG antibody titers in healthcare personnel. PLoS One. 2022 Mar 2;17:e0263942.

Minor comments.

I recomend to rewritte tha abstract more clearly.

Line 27: remove “were assigned” at the end of the sentences.

Line 35, I think the full stop should be replaced with a comma (“gradually declining. While”).

Lines 37-38: “…which were similar to or higher than the geometric mean titer of a panel of SARS-CoV-2 convalescent serum samples.” I cannot find these results in the text.

Lines 94-95: “The titer of neutralizing antibodies was determined with a CLIA-based assay MAGLUMI SARS-CoV-2 neutralizing antibody detection kit (Snibe Diagnostic Shenzhen, China). “ In my opinion also performing a pseudovirus neutralizing assay would have been very suitable test to assess the immune response.

Lines 163-164. Please, rewrite in a clearer way.

Lines 168-170 and Table 2. Please, explain in Material and method s what measured the variables “Time x treatment”, “Time”, “Treatment” and “Subject”.

Limitations. The authors should include the limitations of their study.

Round 2

Reviewer 1 Report

Noted that the authors have tried to respond the the previous comments. 

The description of the spike protein has been mentioned, so that is fine. 

The normalization needs to be explained better, not just simply stating that they were normalized. Was it a percentage normalization and on what parameter? These details are important for better comparisons. This should be made clearer before publication. 

Apart from this, noted that other changes have been effected and is fine. 

Author Response

We thank the reviewer's comment. We have included the unnormalized raw data of the neutralizing antibody level (in µg/mL) apart from the normalized data. We have also discussed on the data in the updated manuscript.

Reviewer 2 Report

The manuscript can be accepted for publication in its current form.

Author Response

We thank the reviewer's positive comments.

Reviewer 4 Report

1.  Although I understand (and appreciate) the authors´ explanations, in my opinion the normalization of nAb to day 1 is a relative measure and might be very high or very low. I think that nAb levels should have been expressed in an objective way (ideally in Binding Antibody Units (BAU) or at least in arbitrary units (AU/mL).

In fact, nAb were measured using a CLIA-based assay, so these data are available to authors. That way the reader will be able to see the quantification of the immunoreactivity (which could be non-significant using absolute and no relative units), especially when there is so much variation on nAB titres depending on the persons, vaccine type, time since infection or vaccination, …

The graphic could be replaced with (for example) another plot showing the means, boxes with the interquartile range or standard deviation, whiskers to indicate the spread of the data,...

2. After 8 months of a second vaccine dose (Day 1), the level of neutralizing antibodies varies a lot depending on the type of vaccine used (inactivated virus or mRNA). A separate analysis of the booster effect (measured in units) with the heterologous vaccine should have been done for each of the previous vaccines used. At least, normalization at day 1 should have been done independently for each type of previous vaccine.

3. Although there were not statistical differences in time from 2nd dose to day 1between vaccinated and placebo groups, what is the reason to select day 1 as such? The authors should explain this issue.

Author Response

  1. Although I understand (and appreciate) the authors´ explanations, in my opinion the normalization of nAb to day 1 is a relative measure and might be very high or very low. I think that nAb levels should have been expressed in an objective way (ideally in Binding Antibody Units (BAU) or at least in arbitrary units (AU/mL).

In fact, nAb were measured using a CLIA-based assay, so these data are available to authors. That way the reader will be able to see the quantification of the immunoreactivity (which could be non-significant using absolute and no relative units), especially when there is so much variation on nAB titres depending on the persons, vaccine type, time since infection or vaccination, …

The graphic could be replaced with (for example) another plot showing the means, boxes with the interquartile range or standard deviation, whiskers to indicate the spread of the data,...

Thank you for your comment. We have included the neutralizing antibody level before normalization in Table 1. However, since the level of neutralizing antibody varied tremendously (e.g. Day 1 from 0.036 to 1.030 mg/ml, it may not be meaningful to plot the data without normalization.

  1. After 8 months of a second vaccine dose (Day 1), the level of neutralizing antibodies varies a lot depending on the type of vaccine used (inactivated virus or mRNA). A separate analysis of the booster effect (measured in units) with the heterologous vaccine should have been done for each of the previous vaccines used.At least, normalization at day 1 should have been done independently for each type of previous vaccine.

Thank you for your comment. We have treated the data separately for the previous vaccines used in Table 3. Except on Day 15 where significant different was observed in the booster group for BNT162b2 (n=3) and BBIBP (n=6), there was no significant difference between other data (one-way ANOVA, p > 0.05) on the same day under the same treatment group. Such statistical difference on Day 15 of the booster group may be caused by small sample size, since no significant different when considering all participants that previous treated with BNT162b2 (n = 5; 0.935 ± 0.081) and BBIBP (n = 11; 0.732 ± 0.109) on the same day.  To minimize the bias due to small sample size, we thus considered the data combined both previous vaccines treated.

  1. Although there were not statistical differences in time from 2nddose to day 1between vaccinated and placebo groups, what is the reason to select day 1 as such? The authors should explain this issue.

Thank you for your comment. Several studies reported that the neutralizing antibody level declined over time and reached a stabilization phase at low levels (~ 60% for BNT162b2 and 15% for BBIBP) from 6 months after the 2nd dose of the primary vaccination cycle. The starting date of the booster study was thus chosen using at least 6 months as the criteria. We have added this point in Section 3.1 to explain this.
